# Productive and Qualitative Characteristics of Pasture and Performance of Sheep Grazing on *Urochloa* Cultivars Under Intermittent Stocking

**DOI:** 10.3390/ani15121783

**Published:** 2025-06-17

**Authors:** Rodrigo da Silva Santos, João Virgínio Emerenciano Neto, Stela Antas Urbano, Francisco Israel Lopes Sousa, Anne Carolinne Rodrigues Linhares da Silva, Maria Alice de Lima Soares, Ana Beatriz Graciano da Costa, Juliana Caroline Santos Santana, Antonio Leandro Chaves Gurgel, Marcone Geraldo Costa, Gelson dos Santos Difante

**Affiliations:** 1Graduate Program in Animal Science, Federal University of Vale do São Francisco, Petrolina 56304-917, Pernambuco, Brazil; rodrigosilva1509@gmail.com (R.d.S.S.); israellopes20@hotmail.com (F.I.L.S.); beatrizcosta.0303@hotmail.com (A.B.G.d.C.); 2Academic Unit Specialized in Agrarian Sciences, Federal University of Rio Grande do Norte, Macaíba 59280-000, Rio Grande do Norte, Brazil; stela_antas@yahoo.com.br (S.A.U.); annecarolinnelinhares@gmail.com (A.C.R.L.d.S.); mariaalicesoares07@gmail.com (M.A.d.L.S.); jukrol_@hotmail.com (J.C.S.S.); marcone.costa@ufrn.br (M.G.C.); 3Campus Professora Cinobelina Elvas, Federal University of Piauí, Bom Jesus 64900-000, Piauí, Brazil; antonioleandro09@gmail.com; 4College of Veterinary Medicine and Animal Science, Federal University of Mato Grosso do Sul, Campo Grande 79070-900, Mato Grosso do Sul, Brazil; gdifante@hotmail.com

**Keywords:** animal performance, forage mass, gain per unit area, grazing behavior Ipyporã, nutrition, tropical pastures

## Abstract

Most sheep meat production systems are based on the exploitation of native pastures. This practice results in low food supply and reduced animal performance. The use of cultivated pastures can significantly contribute to improving zootechnical performance. In this study, we assessed the production potential, nutritional quality, and adaptation of these pastures to tropical environments, as well as the performance and ingestive behavior of grazing animals. We found that the Ipyporã cultivar outperformed the other evaluated cultivars in forage production, with an increase of 1125.37 kg DM/ha compared to the average of the other cultivars, and consequently, enhanced sheep meat production per hectare. *Urochloa* cultivars are suitable for diversifying pasture systems.

## 1. Introduction

The Brazilian livestock production is firmly grounded in the raising of ruminants on pasture-based systems [1]. In the Northeast region, these production systems are typically based on native pastures, often without proper management practices for efficient production [2,3]. Therefore, using cultivated pastures is a viable strategy to enhance herd productivity and agricultural profitability [4,5].

Sheep production in systems based on tropical pastures has shown satisfactory results [6], emphasizing the importance of forage grasses. However, extrapolating results from different environmental conditions may cause economic losses and issues, such as pasture degradation. Therefore, evaluating the response of forage grasses to sheep grazing in the Brazilian Northeast is essential to support management adaptations, especially regarding the use of commercially available forage species and appropriate adjustments in forage management practices, given the knowledge gap caused by the predominance of traditional feeding systems based on native Caatinga vegetation, which is characterized by xerophilous plants adapted to dry climates and water scarcity.

Species of the *Urochloa* genus (syn. *Brachiaria*) stand out for their adaptability to a wide range of environments, high forage production, and ease of management [7], which justifies their widespread adoption across Brazil. Moreover, the genetic potential of this genus has enabled the continual release of new cultivars through plant breeding programs, yielding positive results for livestock production in the country. These advances will further enhance plant and animal productivity across various ecosystems.

Although *Urochloa* species are widely distributed, *U. brizantha* cv. Marandu remains predominant. Released in 1984 as an alternative to monocultures of *U. decumbens*, Marandu gained popularity due to its resistance to spittlebugs. Grazing trials have shown that *U. brizantha* cvs. BRS Paiaguás, and Xaraés, as well as the hybrid *U. brizantha × U. ruziziensis* cv. Ipyporã yields satisfactory results in terms of animal performance [8,9,10]. These findings demonstrate the feasibility of using these cultivars as the primary feed base in grazing systems. Diversifying pasture areas is a fundamental principle for the sustainability of production systems, which supports the need to evaluate new *Urochloa* cultivars, particularly in Brazil Northeast, where the sheep population and mutton consumption are highest [11].

We hypothesize that *Urochloa* cultivars can enhance animal productivity and reduce reliance on cv. Marandu monocultures in the Brazilian Northeast. In this context, the objective of this study was to evaluate the productive and qualitative characteristics of the pasture, as well as the performance of sheep grazing on *Urochloa* cultivars managed under intermittent stocking.

## 2. Materials and Methods

All experimental procedures described in this study were reviewed and approved by the Animal Ethics Committee for Teaching and Research of the Federal University of Rio Grande do Norte (CEUA-UFRN) under protocol number 029/2021.

### 2.1. Experimental Site and Period

The experiment was conducted at the Specialized Academic Unit in Agricultural Sciences, on the campus of the Federal University of Rio Grande do Norte (UFRN), in Macaíba, RN, in the research area of the Study Group on Forage and Ruminant Production (GEFORP), located at 5°53′34″ S and 35°21′50″ W, from March to September 2022 (rainy season). The experimental period was conducted from March to September 2022, totaling 180 days.

According to the Köppen classification, the region’s climate is a transition between types As’ and BSh’, with an average annual temperature of 27.1 °C and an average annual rainfall of 1058.1 mm [12]. During the experimental period, the average minimum and maximum temperatures were 24.6 °C and 25.7 °C, respectively, with a total rainfall of 1210 mm (Figure 1).

### 2.2. Design and Conduct of the Trial

A completely randomized design was used, with six replications. Treatments consisted of four forage grass cultivars: *Urochloa brizantha* cvs. Marandu, Xaraés, and Paiaguás, and the hybrid *U. ruziziensis × U. brizantha* cv. BRS Ipyporã. Each experimental unit measured 600 m^2^ (40 × 15 m) and was subdivided into six paddocks per cultivar, totaling 24 units. The total area used for the experiment was 1.44 ha (14,400 m^2^). Pastures were managed under intermittent stocking, with grazing initiated when the canopy reached 40 cm in height (pre-grazing target) and removed when it was reduced to 20 cm (post-grazing target).

The soil in the experimental area was classified as *Arenosols*, corresponding to Entisols (Quartzipsamments) soil taxonomy [13]. Before pasture establishment, soil samples were collected from the 0–20 cm layer and submitted for physical and chemical characterization (Table 1).

The soil was conventionally prepared using plowing and harrowing. Sowing was performed by broadcasting, considering the seed purity and germination rate (cultural value, VC%) and the recommended seeding rate for each cultivar. Fertilization was applied at sowing by incorporating 25 kg/ha of P_2_O_5_ (as single superphosphate) into the soil. Topdressing was carried out with 40 kg/ha of K_2_O (potassium chloride) and 50 kg/ha of N (urea) after the second grazing cycle of each paddock.

### 2.3. Pasture Evaluations

Pasture evaluations were conducted before and after grazing. Canopy height was measured using a graduated ruler in centimeters (20 measurements per paddock), taken from ground level to the “leaf horizon” at the top of the canopy. Light interception (LI) and leaf area index (LAI) measurements were taken at 10 distinct points within each paddock using a Ceptometer model Accupar LP-80^®^ of METER group, Pullman, WA, USA.

Using a manual mower, three samples were collected to estimate forage mass by cutting at ground level (0.5 × 1.0 m) within each experimental unit at random sampling points. The samples were separated into leaf blade, stem (stem + sheath), and dead material fractions. These were then weighed and placed in a forced-air drying oven at 55 °C for 72 h to determine the dry mass, percentage of morphological components, and the leaf blade-to-stem ratio.

Composite forage samples of the morphological components (leaf blade and stem) were prepared for chemical analysis. These samples were then ground using a Wiley-type knife mill (Tecnal, Piracicaba, São Paulo, Brazil) fitted with a 1 mm sieve to determine forage chemical composition. The contents of dry matter (DM), ash, crude protein, neutral detergent fiber (NDF), acid detergent fiber (ADF), and lignin (LIG) were determined according to the methodologies described by AOAC [14].

Forage allowance was calculated following the methodology proposed by Sollenberger et al. [15], taking into account leaf accumulation during the grazing period. Forage disappearance during grazing was calculated as the difference between pre- and post-grazing forage mass within the same grazing cycle. Disappearance values were then expressed as a percentage of the pre-grazing forage mass.

### 2.4. Animal Performance Evaluations

Santa Inês sheep were used in the study, six animals (replicates) per cultivar, totaling twenty-four weaned and castrated male lambs, with an initial average live weight of 22.2 ± 3.01 kg and an average age of 83 days. The lambs were randomly assigned to the different forage cultivars and managed under an intermittent stocking system with variable stocking rates. Animals were monitored until they reached a live weight of 32 kg.

Before the experimental period, the animals were vaccinated against clostridial diseases and treated for both endo- and ectoparasites. Parasite control was performed using the FAMACHA^©^ method, which guided the administration of anthelmintics [16]. The lambs were kept on pasture during the day (from 8:00 a.m. to 4:00 p.m.) and housed in collective pens by the treatment group at dusk for overnight shelter. Water was provided *ad libitum* both in the pens and in the paddocks.

The animals received a daily concentrate supplement equivalent to 1% of their live weight in the pens. The concentrate comprised 75.5% ground corn, 20% soybean meal, 1.0% mineral salt, 2.5% urea + ammonium sulfate (9:1), and 1.0% sodium chloride, which was provided in private troughs at 3:00 PM.

The formulation was based on the nutritional requirements recommended by the NRC [17] to support an average daily gain of approximately 80 g/animal. Samples of the concentrate ingredients, ground corn, and soybean meal were submitted for chemical composition analysis (Table 2), following the methodologies described by AOAC [14].

Test animals were weighed every seven days, while regulators were weighed monthly to monitor weight gain and adjust the stocking rate. Average daily gain (ADG) was calculated by dividing the difference in body weight between weekly measurements by seven. Total weight gain (TWG) was determined as the difference between the final and initial weights of the test animals. Animal weight gain per hectare was obtained by multiplying the ADG of the test animals by the number of animals maintained per hectare in each grazing cycle. The stocking rate was adjusted weekly based on forage mass, with a minimum of six animals maintained per paddock.

Grazing time is calculated as how period, expressed in days, during which the animals remain in the paddock until the condition target for post-grazing.

Rest period is the interval, also expressed in days, between the animals’ exit from the paddock and their subsequent return.

Ingestive behavior was evaluated over a continuous 48 h period, conducted simultaneously across all treatments. All animals from each treatment were identified with different colored markers to facilitate individual recognition and were observed by trained evaluators. The grazing, rumination, and idling activities of each animal were recorded at five-minute intervals throughout the observation period. Additionally, while the animals were on pasture, the bite rate was measured by recording the time required for each animal to perform 20 consecutive bites; this procedure was repeated six times per animal, distributed across different periods of the day during grazing. The bite rate was then expressed as the number of bites per minute [18].

### 2.5. Statistical Analysis

Data were subjected to analysis of variance, and when significant, treatment means were compared using the Tukey test at a 5% significance level, performed with the statistical software SISVAR, version 5.6 [19]. The following statistical model was used: Yij = μ + Ci + βij, where Yij = observed value of cultivar i in replicate j; μ = overall mean effect; Ci = effect of cultivar i (i = Marandu, Paiaguás, Xaraés, and Ipyporã); and βij = random error associated with the plot of cultivar i in replicate j.

## 3. Results

There were no significant differences in grazing period (*p* = 0.3513) or rest period (*p* = 0.3023) among the cultivars, with average values of 9.9 and 43.5 days, respectively. Pre-grazing canopy height also did not differ among cultivars (*p* = 0.2093; Table 3), with a mean of 43.6 cm. No differences were observed for light interception (LI) (*p* = 0.5024) or leaf area index (LAI) (*p* = 0.1312) before grazing, with mean values of 80.3% and 2.35, respectively.

The highest pre-grazing forage mass (*p* = 0.0047) and stem mass (*p* = 0.0053) were observed in cv. Ipyporã, while the other cultivars did not differ from each other. Leaf blade mass was also higher in cv. Ipyporã (*p* = 0.0335); Marandu and Xaraés cultivars showed intermediate values, with an average of 1664.7 kg DM/ha, whereas Paiaguás had the lowest value.

A greater amount of dead material was observed in cv. Ipyporã (*p* = 0.0123), while the lowest values were recorded for cvs. Xaraés and Paiaguás. The highest leaf blade-to-stem ratio was found in cv. Xaraés (*p* = 0.0453), whereas the lowest ratio was observed in cv. Paiaguás.

Regarding post-grazing productive characteristics, canopy height was greater (*p* = 0.0008) in cvs. Paiaguás and Ipyporã, while lower values were observed in the cvs. Xaraés and Marandu (Table 4).

Post-grazing light interception (LI) (*p* = 0.1795), leaf area index (LAI) (*p* = 0.2683), leaf blade mass (*p* = 0.8713), stem mass (*p* = 0.2064), and leaf blade-to-stem ratio (*p* = 0.4483) did not differ among cultivars, with mean values of 40%, 0.68, 443.2 kg DM/ha, 982.6 kg DM/ha, and 0.45, respectively. The Paiaguás cultivar showed the lowest forage mass (*p* = 0.0071) and dead material mass (*p* = 0.0007).

The contents of ash, crude protein, and acid detergent fiber (ADF) in both leaf blades and stems of the cultivars of *Urochloa* were similar (*p* > 0.05) (Table 5). Neutral detergent fiber (NDF) content in the stem was also not affected by cultivar (*p* > 0.05).

Neutral detergent fiber (NDF) content in the leaf blade was higher (*p* = 0.0062) in cvs. Xaraés and Ipyporã, with the lowest value observed in cv. Marandu. Cultivars Marandu, Paiaguás, and Ipyporã exhibited lignin content approximately 1.46 times higher than cv. Xaraés, representing a difference of 2.15% of the dry matter.

Regarding lignin content in the stem, cvs. Marandu and Paiaguás showed the highest values, with an average of 10.71% of dry matter, while cv. Xaraés presented the lowest observed value.

Regarding ingestive behavior and sheep performance (Table 6), no significant differences were observed among cultivars for forage disappearance (*p* > 0.05), idling time (*p* = 0.2255), rumination time (*p* > 0.05), bite rate (*p* > 0.05), total weight gain (*p* = 0.9397), or average daily gain (*p* = 0.9397), with mean values of 39.15%; 592.1 min/day; 466.9 min/day; 19.95 bites/min; 9.43 kg/animal; and 74.6 g/animal/day, respectively.

Grazing time was longer (*p* = 0.0048) in cv. Paiaguás and shorter in cvs. Marandu and Xaraés. Leaf allowance was higher (*p* < 0.05) in cvs. Xaraés and Paiaguás, with an average of 6.46 kg DM/100 kg LW. Stocking rate (SR) (*p* < 0.0001) and average daily gain per area (ADG/ha) (*p* = 0.0006) were higher in cv. Ipyporã compared to the other cultivars, which did not differ from each other. The Ipyporã cultivar showed a stocking rate twice as high as the other cultivars, averaging 8.79 AU 30 kg/ha. The ADG/ha of animals grazing on cv. Ipyporã was 1.72 times greater than the other cultivars, averaging 769.30 g/ha.

## 4. Discussion

The canopy heights of the *Urochloa* cultivars remained close to the target range recommended for pre-grazing throughout the experiment (43.6 ± 1.44 cm), with no variation among cultivars. The uniformity in height ensured similar shading levels across cultivars, resulting in comparable values for light interception (LI) and leaf area index (LAI). Under intermittent stocking, *Urochloa* cultivars grazed by sheep with a pre-grazing height of 40 cm exhibited an LI of approximately 80%.

Although this LI was below the threshold considered ideal for suppressing pasture regrowth (95% LI), achieving such a level would require greater leaf biomass capable of intercepting more light, which in turn would demand a longer rest period [20] and, consequently, a taller canopy.

In general, small ruminants are more selective grazers. They require more energy relative to their intestinal capacity, and their oral anatomy allows them to graze more selectively and closer to the ground than large ruminants. This selectivity enables them to choose forage of higher nutritional quality [21]. Furthermore, because of their shorter stature, applying pasture height recommendations developed for cattle is inappropriate for sheep. In some situations, the forage canopy could exceed the animals’ height, hindering access to forage and compromising adequate intake. Under the same environmental and management conditions, *Urochloa* cultivars exhibit similar LAI, this study observed an average LAI of 2.35 (Table 4).

Since the pastures were managed at the same pre-grazing height (43.6 cm; Table 3), forage mass, leaf blade mass, and stem mass depended on each cultivar’s structural characteristics. The Paiaguás cultivar has narrower and shorter leaves [22], which may have contributed to its lower leaf blade mass.

On the other hand, cv. Ipyporã exhibits structural traits similar to those of the more productive cultivars studied, such as Marandu and Xaraés, forming a more prostrate and denser sward, which may explain the higher forage and stem mass observed in Ipyporã compared to the other cultivars. Another crucial point to highlight is that all cultivars exhibited a greater amount of leaf blade mass than the other components of the canopy structure (stem and dead material). This structural condition favors intake, as leaves contain less fiber and more protein, making them the preferred fraction for ruminants and enhancing their ability to convert forage into animal products.

Another key component in pasture management assessment is the mass of dead material, representing forage losses from biomass produced but not harvested by the animals [23]. Thus, lowering the pre-grazing height can be a management strategy to reduce the proportion of this component in the total forage mass, especially in the case of cv. Ipyporã, which showed a high accumulation of dead material. According to Echeverria et al. [24], cutting or grazing of cv. Ipyporã is recommended when the canopy reaches approximately 30 cm in height. The 40 cm pre-grazing height adopted for cv. Ipyporã in this study led to greater stem elongation and forage senescence. Nonetheless, under these conditions, this cultivar still presented a high leaf blade-to-stem ratio (LBSR) (1.24, Table 3).

All cultivars exhibited a LBSR greater than 1, indicating that animals had greater access to leaf material during grazing. A ratio of 1 is considered critical in grazing environments, as values below this threshold reflect lower forage quality [25] due to higher accumulation of stems and senescent material. A similar response was reported by Oliveira et al. [26] who evaluated three cutting frequencies (30, 45, and 60 days) in five *Urochloa* cultivars (Basilisk, Marandu, Mulato I, Piatã, and Xaraés) and found that at a 45-day regrowth interval, the cultivars did not differ for this variable; an average LBSR of 1.41 was obtained, a value comparable to that observed in the present study under a similar rest period of 43.5 days.

Post-grazing pasture heights corresponded to 52.61% of the pre-grazing height, a level considered ideal for regrowth interruption. Values below this threshold lead to a marked decrease in the number of bites taken by the animals due to the increased proportion of stems and senescent material, which are more densely distributed in the lower canopy strata. Euclides et al. [27] observed that post-grazing heights representing 33.33% of pre-grazing height, as opposed to 55.56%, resulted in reduced average daily gain due to a lower proportion of leaves, a reduced leaf-to-stem ratio, and higher percentages of stem and dead material.

Similarly, the post-grazing light interception was equivalent to 50% of the pre-grazing light interception (80.1% vs. 40%), a result that correlates with canopy height. Since forage disappearance did not differ among cultivars (Table 6), post-grazing forage mass followed the same pattern as pre-grazing forage and leaf blade mass, which explains the lower value observed in cv. Paiaguás, an outcome attributed to the structural characteristics inherent to this cultivar as previously discussed.

Residual leaf blade mass (post-grazing) is crucial for photosynthesis and pasture regrowth as excessive defoliation reduces regrowth vigor by removing apical meristems [28]. To maintain regrowth capacity, it is recommended that residual height be about 50% of the pre-grazing height; excessive stem mass, which hinders grazing, can be managed by adjusting grazing height or increasing stocking rate to limit stem elongation.

Post-grazing dead material mass was relatively high, representing 43%, 39%, 35%, and 30% of the forage mass in Ipyporã, Xaraés, Marandu, and Paiaguás, respectively (Table 4). The accumulation of senescent material indicates low forage utilization, reducing pasture productivity efficiency [29]. To minimize these losses, shorter regrowth periods or lower pre-grazing heights are recommended. The higher proportion of dead material in Ipyporã and Xaraés suggests that the adopted grazing height may have favored senescent leaf accumulation.

The low post-grazing LBSR was expected as sheep are more selective grazers, primarily consuming leaf blades. For faster pasture regrowth under more frequent grazing cycles, it is recommended to maintain a higher residual LBSR by adopting shorter intervals between grazing events.

The forages evaluated belong to the same genus and had the same regrowth period, explaining the minimal differences in their chemical composition. When testing different regrowth ages, Santana et al. [30] found that these differences did not affect fiber concentration in the leaves but did influence organic matter content and reduced crude protein levels.

These results indicate that fiber content is primarily determined by the genetic potential of each cultivar, whereas forage management can influence mineral and crude protein concentrations. The cultivars Xaraés and Ipyporã exhibited high NDF levels, which may be associated with their greater forage production potential under high rainfall conditions [31,32], as observed in this experiment conducted during the rainy season. Such conditions promote more vigorous growth, leading to a greater accumulation of fibrous material due to increased tissue turnover.

Tropical grasses exposed to high water availability tend to exhibit increased tissue turnover, which can result in greater deposition of fibrous material [33]. The cultivar Xaraés is known for its high forage production potential, especially under conditions of high precipitation [31], while cv. Ipyporã shows increased leaf appearance and elongation rates under higher rainfall and average temperature conditions [32]. Therefore, it can be inferred that favorable environmental conditions, combined with the intrinsic characteristics of Xaraés and Ipyporã cultivars, promote more vigorous growth, resulting in greater fibrous material deposition, primarily hemicellulose in the cell wall and consequently higher NDF content.

A similar result for crude protein was reported by Maia et al. [34] who evaluated the yield and composition of *Urochloa* cultivars during the dry season and found no difference in crude protein content between cvs. Xaraés and Marandu during the rainy period. Carvalho et al. [35], assessing the responses of *Urochloa* cultivars under grazing, observed an average crude protein content of 12.85% in cv. Ipyporã, which was higher than that found in the present study. However, the difference may be attributed to using less developed pastures (30 cm in height), which could explain the higher protein content.

The lower leaf allowance observed in cv. Ipyporã may be attributed to the higher stocking rate for this cultivar (Table 6). Leaf allowance across all cultivars can be considered satisfactory as lambs weighing approximately 30 kg have an average daily dry matter intake of around 3.5% of their live weight [36], indicating that sufficient forage was available. It is important to note that increasing the stocking rate to reduce allowance to levels very close to the recommended threshold (3.5 kg DM/100 kg LW per day) may negatively affect pasture residue structure and, consequently, regrowth rate. Therefore, increases in stocking rates should be guided by the targeted pre- and post-grazing heights rather than solely by forage allowance.

Grazing time is influenced by pasture structure, such as forage mass, height, and LBSR. In this study, grazing time ranged from 22.9% to 25.7% of total activity, being lower than time spent ruminating or idling. The restricted grazing period (8:00 a.m. to 4:00 p.m.) likely contributed to this result as sheep typically show diurnal grazing peaks at sunrise and late afternoon [37]. Similarly, Emerenciano Neto et al. [38] reported that sheep utilize all available grazing time in *Urochloa* and *Megathyrsus* pastures.

Idling was the activity to which the sheep dedicated the largest portion of their time during the day (41.1%), which can be explained by the management system, in which the animals spent more time in the pens (16 h/day) and less time on pasture (8 h/day). Rumination time is influenced by the nature of the diet, particularly the characteristics of the forage. According to Van Soest [39], the higher the fiber content, the longer the rumination period, whereas diets with higher concentrate levels result in shorter rumination periods.

The average bite rate (19.95 bites/min) is consistent with the findings of Silva et al. [29], who reported values ranging from 19.4 to 27.7 bites/min in *Urochloa brizantha* cvs. Marandu and Piatã pastures grazed by sheep. According to Fonseca et al. [40], bite rate varies depending on pasture structure; lower forages with higher stem proportions make forage capture more difficult for the animals.

The lack of significant effects on total weight gain and average daily gain (ADG) may be partially explained by the similar crude protein content across cultivars. The ADG observed in this study was higher than the 40, 50, and 60 g/animal/day reported by Silva et al. [10] for *Urochloa* cvs. Piatã, Xaraés, and Paiaguás, respectively, although those results were obtained during the dry season with a concentrate supplementation of 0.8% of body weight. Evaluating sheep performance during the rainy season, Emerenciano Neto et al. [1] reported ADG values of 53.6 and 40.2 g/animal for cvs. Marandu and Piatã, respectively, which were lower than the average observed in the present study for the same season. However, the lack of concentrate supplementation in their study may explain this difference.

The higher stocking rate and greater daily gain per area observed for cv. Ipyporã can be attributed to its higher pre-grazing forage mass (Table 3), which provided greater carrying capacity. Comparisons between the results of this study and those in the literature are limited due to the scarcity of research evaluating this grass in meat sheep production systems. However, studies with cattle have provided insights into animal performance on this type of pasture. Euclides et al. [9] observed that cv. Ipyporã resulted in greater individual cattle weight gain due to its higher nutritional value compared to Marandu grass. This contrasts with the present study where crude protein levels did not differ among cultivars. Thus, the higher gain per area observed for cv. Ipyporã in this study can be attributed solely to its greater forage mass.

## 5. Conclusions

The cultivars Paiaguás, Xaraés, and Ipyporã show potential for diversifying cultivated pasture areas, offering viable alternatives to the monoculture of cv. Marandu in ruminant production systems. Cultivars with a higher proportion of leaves increase grazing efficiency, reducing the time animals spend grazing. The cultivar Ipyporã stands out for its higher forage productivity, which results in greater animal productivity during the rainy season under intermittent stocking.

## Figures and Tables

**Figure 1 animals-15-01783-f001:**
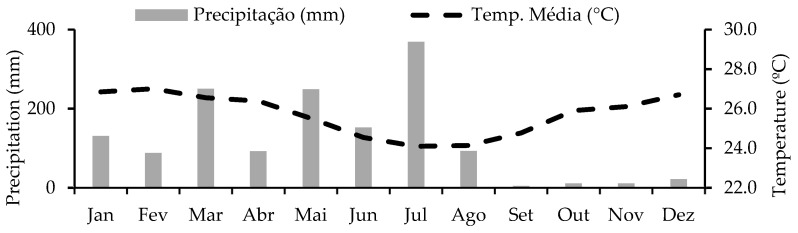
Monthly precipitation and average temperature from January to December 2022.

**Table 1 animals-15-01783-t001:** Chemical and physical characteristics of the soil (0–20 cm layer) in the experimental area.

pH H_2_O	P	K	Na	Ca	Mg	Al	H+Al	CEC	BS	Sand	Silt	Clay
	mg/dm^3^	cmolc/dm^3^							%	g/kg		
5.6	3.16	0.05	0.03	0.5	0.3	0.00	1.16	2.04	44	913	51	36

CEC: cation exchange capacity at pH 7; BS: base saturation (SB/CEC) × 100; SB: sum of bases (Ca + Mg + K).

**Table 2 animals-15-01783-t002:** Chemical composition of the ingredients used in the experimental diet.

Nutritional Components	Ingredients	
	Ground Corn	Soybean Meal
Dry matter (%)	90.28	91.06
Organic matter (% DM)	98.82	92.23
Mineral matter (% DM)	1.18	7.77
Neutral detergent fiber (% DM)	10.34	23.73
Acid detergent fiber (% DM)	2.62	8.51
Hemicellulose (% DM)	7.72	15.23
Lignin (% DM)	0.47	0.05
Crude protein (% DM)	7.49	50.35

DM: Dry matter.

**Table 3 animals-15-01783-t003:** Productive characteristics of the canopy before grazing in four *Urochloa* cultivars grazed by sheep.

Variable	Marandu	Paiaguás	Xaraés	Ipyporã	SEM
Canopy height (cm)	41.6	44.7	43.5	44.6	1.13
Light interception (%)	80.2	80.6	74.9	85.5	3.23
Leaf area index	2.38	2.40	2.11	2.50	0.19
Forage mass (kg DM/ha)	3776.2 b	3215.9 b	3642.7 b	4670.3 a	277.08
Leaf blade mass (kg DM/ha)	1575.8 ab	1327.0 b	1753.7 ab	1842.0 a	156.43
Stem mass (kg DM/ha)	1237.2 b	1225.4 b	1102.3 b	1581.1 a	99.89
Dead material (kg DM/ha)	963.3 ab	663.5 b	786.6 b	1247.2 a	127.16
Leaf blade-to-stem ratio	1.29 ab	1.09 b	1.61 a	1.24 ab	0.13

SEM, standard error of the mean. Means followed by different letters in the row differ according to the Tukey test (*p* < 0.05).

**Table 4 animals-15-01783-t004:** Productive characteristics of the canopy after grazing in four Urochloa cultivars grazed by sheep.

Variable	Marandu	Paiaguás	Xaraés	Ipyporã	SEM
Canopy height (cm)	21.99 b	24.15 a	21.86 b	23.77 a	0.45
Light interception (%)	43.9	30.8	38.6	46.8	5.24
Leaf area index	0.83	0.68	0.67	0.56	0.10
Forage mass (kg DM/ha)	2341.9 a	1804.2 b	2292.8 a	2661.3 a	159.25
Leaf mass (kg DM/ha)	452.0	431.5	417.0	472.3	52.72
Stem mass (kg DM/ha)	1069.8 a	837.8 a	970.2 a	1052.7 a	79.06
Dead material (kg DM/ha)	820.0 b	534.9 c	905.6 ab	1136.3 a	93.08
Leaf blade-to-stem ratio	0.43	0.52	0.47	0.39	0.06

SEM, standard error of the mean. Means followed by different letters differ according to the Tukey test (*p* < 0.05).

**Table 5 animals-15-01783-t005:** Chemical composition of the canopy before grazing in four *Urochloa* cultivars grazed by sheep.

Variables (% of MS)	Marandu	Paiaguás	Xaraés	Ipyporã	SEM
**Leaf blade**					
Ash	8.84	7.61	7.78	7.59	0.56
Neutral detergent fiber	55.56 b	58.52 ab	62.5 a	64.18 a	1.71
Acid detergent fiber	35.03	31.78	33.65	32.62	1.20
Acid detergent lignin	6.72 a	7.34 a	4.66 b	6.37 a	0.54
Crude protein	10.54	9.25	10.24	9.04	1.15
**Stem**					
Ash	7.63	6.70	7.83	7.05	0.71
Neutral detergent fiber	69.35	73.37	71.10	72.79	1.31
Acid detergent fiber	41.54	45.53	41.83	42.43	1.59
Acid detergent lignin	10.06 a	11.36 a	7.46 b	9.53 ab	0.58
Crude protein	5.45	4.55	5.50	4.78	0.88

SEM, standard error of the mean. Means followed by different letters differ according to the Tukey test (*p* < 0.05).

**Table 6 animals-15-01783-t006:** Ingestive behavior and performance of Santa Inês sheep grazing on *Urochloa* pastures.

Variables	Marandu	Paiaguás	Xaraés	Ipyporã	SEM
Forage disappearance (%)	39.3	41.8	38.4	37.1	3.83
Leaf allowance (kg DM/100 kg LW)	5.36 ab	6.76 a	6.16 a	3.93 b	1.28
Grazing time (min/day)	332.5 b	370.8 a	327.5 b	358.3 ab	9.28
Idling time (min/day)	632.5	574.2	560.8	600.8	25.19
Rumination time (min/day)	441.7	443.3	518.3	464.2	20.01
Bite rate (bites/min)	19.50	20.52	20.16	19.63	0.82
Total weight gain (kg/animal)	9.27	9.02	9.95	9.50	1.08
Stocking rate (AU 30 kg/ha)	8.22 b	8.40b	9.74 b	17.59 a	1.06
Average daily gain (g/animal)	76.6	83.2	70.8	68.0	7.66
Average gain per area (g/ha)	650.8 b	865.5 b	791.6 b	1319.5 a	83.63

SEM, standard error of the mean. Means followed by different letters in the row differ according to the Tukey test (*p* < 0.05).

## Data Availability

The original contributions presented in this study are included in the article/Appendix A. Further inquiries can be directed to the corresponding author.

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
