# Peer review of "Productive and Qualitative Characteristics of Pasture and Performance of Sheep Grazing on Urochloa Cultivars Under Intermittent Stocking"

_animals, 2025, doi:10.3390/ani15121783_

Round 1

Reviewer 1 Report

Comments and Suggestions for Authors

Simple Summary

The text is understandable but contains long sentences and weak connectors, which hinders quick reading.

The summary mentions that Ipyporã "stands out," but does not provide concrete data, such as percentage gain or a clear comparison with the other cultivars.

The term 'evaluated' is repeated in close proximity.

The last sentence is vague ("can be used to diversify environments").

Abstract

Indicate which statistical test was used.

Add (P < 0.05) or (P > 0.05) to indicate statistical differences.

956.2 g/ha?

In the conclusion, the authors state that all cultivars were satisfactory, but do not detail which aspects. Highlight the strengths and limitations of each cultivar, even if briefly.

Keywords: using the wordforageis too generic.

There is no need to useingestive behaviorandgrazing behavior”.

Use terms that reflect the article, such as cultivar names, and tropical pastures.

Introduction

Lines 42–55 provide a good overview of the production system but can be summarized to make the text more fluid.

The introduction addresses several important topics (extensive production, lack of management, scarcity of regional data, and the potential of Urochloa grasses). Still, it does not objectively state the scientific gap the study aims to fill.

The current hypothesis is long and more descriptive than scientific.

Methodology

The sampling for evaluating forage mass and morphological components is mentioned but lacks details. Were the sampling points random?

The forage chemical analysis needs more detail, it is unclear whether the analysis was done per paddock, per grazing cycle, or if samples were composited.

Information on the supplement feeding time is missing.

The statistical model is presented, but it is unclear whether basic assumptions (normality, homogeneity of variances) were verified.

Results

Group the results in the order of morphological characteristics, chemical composition, and ingestive behavior, this may result in more fluid reading and more logical transitions.

There is repeated use of terms likethere was no significant differencewithout variation in wording.

Discussion

The discussion contains long and dense passages, with many ideas strung together without clear divisions. The text does not flow well.

Reorganize the discussion to allow a more fluid reading (e.g., pasture structure and management, chemical composition, ingestive behavior, animal performance, pasture use efficiency).

Some practical implications of the results are mentioned without emphasis. Insert more conclusive sentences, indicating practical management recommendations.

There is an excessive focus on studies with cattle, updated references, and more work involving small ruminants, especially in tropical systems.

At some points, the discussion mentions values without clearly indicating which table they belong to, making it difficult to verify the data.

Conclusion

There are no practical indications for the use of the cultivars, such as the situations or production systems in which each cultivar would be more advantageous.

Author Response

We appreciate the reviewers' recommendations for improving the article. We have made the necessary corrections to meet the requested requirements, and we have also explained why we were unable to follow some of the suggested advice.

Simple Summary

Comments 1: The text is understandable but contains long sentences and weak connectors, which hinders quick reading.

Response 1: We adjusted the text to improve the flow of reading.

Comments 2: The summary mentions that Ipyporã "stands out," but does not provide concrete data, such as percentage gain or a clear comparison with the other cultivars.

Response 2: Data was added that allows the reader to understand why Ipypora stands out.

Comments 3: The term 'evaluated' is repeated in close proximity.

Response 3: The terms were changed.

Comments 4: The last sentence is vague ("can be used to diversify environments").

Response 4: The sentence was modified to represent the real recommendation that the authors wanted to express.

Abstract

Comments 5: Indicate which statistical test was used.

Comments 6: Add (P < 0.05) or (P > 0.05) to indicate statistical differences.

Reponse 5 and 6:

We understand the importance of statistical information; however, the character limit imposed by the journal for the summary section restricts the inclusion of such details, which are already implicit in the context. We chose to prioritize the inclusion of key data to engage readers and encourage them to consult the full text for methodological specifics.

Comments 7: 956.2 g/ha?

Response 7:

We apologise for the error, and it has been corrected.

Comments 8: In the conclusion, the authors state that all cultivars were satisfactory, but do not detail which aspects. Highlight the strengths and limitations of each cultivar, even if briefly.

Response 8:

We appreciate the comment. Since none of the evaluated variables presented negative results for the cultivars, we conclude that all can be used in production systems. However, we emphasize that, due to its higher productivity, the Ipyporã cultivar is the most suitable, mainly because of its superior forage yield and, consequently, greater potential for sheep meat production per hectare.

Comments 9: Keywords: using the word “forage” is too generic.

There is no need to use “ingestive behavior” and “grazing behavior”.

Use terms that reflect the article, such as cultivar names, and tropical pastures.

Response 9: Thank you for the suggestion. After reviewing the comments, we have reformulated the keywords.

Introduction

Comments 10: Lines 42–55 provide a good overview of the production system but can be summarized to make the text more fluid.

Response 10: Thank you for your consideration. We have made changes to the excerpt to make the text more objective.

Comments 11: The introduction addresses several important topics (extensive production, lack of management, scarcity of regional data, and the potential of Urochloa grasses). Still, it does not objectively state the scientific gap the study aims to fill.

Response 11: The text has been improved to include this information.

Comments 12: The current hypothesis is long and more descriptive than scientific.

Response 12: We appreciate your observation and have modified the hypothesis to improve its flow and make it more objective.

Methodology

Comments 13: The sampling for evaluating forage mass and morphological components is mentioned but lacks details. Were the sampling points random?

Response 13: This information has been added.

Comments 14:The forage chemical analysis needs more detail, it is unclear whether the analysis was done per paddock, per grazing cycle, or if samples were composited.

Response 14: We have added this information.

Comments 15: Information on the supplement feeding time is missing.

Response 15: We have added this information.

Comments 16: The statistical model is presented, but it is unclear whether basic assumptions (normality, homogeneity of variances) were verified.

Response 16: Yes, the variables analyzed in this experiment are well known in the literature to follow a normal distribution; therefore, we have not included this information in the text. For example, in the studies by da Silva et al. (2024). However, if the reviewers consider it essential, we can add this information to the manuscript.

Reference: da Silva, RF; Ribeiro, PHC; Silva, YdS; Soares, M.A.d.L.; Ribeiro, CVDM; Rangel, AHdN; Ferreira, M.d.A.; Emerenciano Neto, J.V.; Urbano, S.A. Development and weight growth curves of Santa Inês ewes on pasture supplemented with concentrate in the pre-weaning phase. Animals 2024, 14, 1766. https://doi.org/10.3390/ani14121766

Results

Comments 17: Group the results in the order of morphological characteristics, chemical composition, and ingestive behavior, this may result in more fluid reading and more logical transitions.

Response 17: We apologize to the reviewer because we did not fully understand the suggestion, as the data are already organised in this exact sequence.

Comments 18: There is repeated use of terms like “there was no significant difference” without variation in wording.

Response 18:  We thank the reviewer for the observation. We have revised the text to avoid the repeated use of the phrase.

Discussion

Comments 19: The discussion contains long and dense passages, with many ideas strung together without clear divisions. The text does not flow well.

Response 19: Thank you for the observation. We have revised the discussion.  We leave more clear, more focused sections to improve text flow.

Comments 20: Reorganize the discussion to allow a more fluid reading (e.g., pasture structure and management, chemical composition, ingestive behavior, animal performance, pasture use efficiency).

Response 20: Some discussions interact with other points, such as forage production affecting animal production, but we have tried to make the text more fluid with modifications.

Comments 21: Some practical implications of the results are mentioned without emphasis. Insert more conclusive sentences, indicating practical management recommendations.

Response 21: Practical implications and management recommendations have been added and emphasized throughout the discussion to better highlight the application of the results.

Comments 22: There is an excessive focus on studies with cattle, updated references, and more work involving small ruminants, especially in tropical systems.

Response 22: Reducing the comparison with cattle and we highlight destacamos the paper with ovines.

Comments 23: At some points, the discussion mentions values without clearly indicating which table they belong to, making it difficult to verify the data.

Response 23: We have improved clarity by explicitly citing the corresponding tables when presenting values.

Conclusion

Comments 24: There are no practical indications for the use of the cultivars, such as the situations or production systems in which each cultivar would be more advantageous.

Response 24: We appreciate the comment. We have made adjustments to meet the recommendation as far as possible. However, we did not extrapolate the results obtained. Our recommendations are restricted to the specific conditions evaluated in this study, under intermittent stocking during the rainy season.

Reviewer 2 Report

Comments and Suggestions for Authors

Some of the corrections in the text are required

Line 81 All to all

Figure 1. Precipitacao to Precipitation

Temp media to ambient temperature

Abbreviation of months in figure need correction Jan, Feb, Mar, Apr, May, June, July, Aug, Sept, Oct, Nov, Dec.

Table H2O - 2 just below the line of text

Table 5 ash to Ash

Abbreviations 

NRC - National Research Council

Author Response

We appreciate the reviewers' recommendations for improving the article. We have made the necessary corrections to meet the requested requirements, and we have also explained why we were unable to follow some of the suggested advice.

Comments 1: Some of the corrections in the text are required

Line 81 All to all

Figure 1. Precipitacao to Precipitation

Temp media to ambient temperature

Abbreviation of months in figure need correction Jan, Feb, Mar, Apr, May, June, July, Aug, Sept, Oct, Nov, Dec.

Table H2O - 2 just below the line of text

Table 5 ash to Ash

Abbreviations 

NRC - National Research Council

Response 1: We apologize for the errors and have fixed them all.

Reviewer 3 Report

Comments and Suggestions for Authors

General review

The topic and the quality of this paper is suitable for Animals Journal. The grazing and pasture management is important in the ruminant species, especially small ruminants. The green forage plants are one of the interests in ruminants’ nutrition.

Detailed review

Title: I think this phrase „intermittent stocking” is not a good choice. Recommended: “… under short period”

Introduction:

This section contains the necessary information about pasture utilisation and grazing management in Brazil.

line 53: please give more info about “Catanga” vegetation!!

Material and methods:

Figure 1: please use only English words!

line 108: Please use The World Reference Base (WRB) classification for soils.

Table 1: BS, % please add more info into footnote!

lines 174-178: this subsection is not clear!! Please explain how to record the behavioural elements! And please explain how and when (and how many times) perform the 20 bites?

Results and Discussion sections are well-prepared.

line 187: How calculated the grazing time (9.9 days) and rest period (43.5 days)!

Conclusions are appropriate and accurate.

Comments on the Quality of English Language

Please pay more attention to the use of professional terms!

Author Response

We appreciate the reviewers' recommendations for improving the article. We have made the necessary corrections to meet the requested requirements, and we have also explained why we were unable to follow some of the suggested advice.

Comments 1: General review

The topic and the quality of this paper is suitable for Animals Journal. The grazing and pasture management is important in the ruminant species, especially small ruminants. The green forage plants are one of the interests in ruminants’ nutrition.

Response 1: We appreciate the recognition of the relevance and quality of our work for the journal Animals, especially on such an important topic as pasture management and the nutrition of small ruminants.

Detailed review

Comments 2: Title: I think this phrase „intermittent stocking” is not a good choice. Recommended: “… under short period”

Response 2: Thank you for the recommendation, we use the term most accepted in the scientific community and also used in articles in the animal journal.

Introduction:

This section contains the necessary information about pasture utilisation and grazing management in Brazil.

Comments 3: line 53: please give more info about “Catanga” vegetation!!

Response 3: Thank you for your suggestion. We have modified the sentence to provide more information about the native Caatinga vegetation, highlighting its main characteristics.

Material and methods:

Comments 4: Figure 1: please use only English words!

Response 4: Figure 2 is now written entirely in English

Comments 5: line 108: Please use The World Reference Base (WRB) classification for soils.

Response 5: We made the exchange as requested

Comments 6: Table 1: BS, % please add more info into footnote!

Response 6: More information about calculating base saturation has been added.

Comments 7: lines 174-178: this subsection is not clear!! Please explain how to record the behavioural elements! And please explain how and when (and how many times) perform the 20 bites?

Response 7: Thank you for your comment. We have revised to clarify the methodology. The description now explicitly details how the behavioral elements were recorded, how the animals were identified, and how and when the bite rate measurements were conducted.

Results and Discussion sections are well-prepared.

Comments 8: line 187: How calculated the grazing time (9.9 days) and rest period (43.5 days)!

Response 8: Is this way: Grazing time refers to the period, expressed in days, during which the animals remain in the paddock until the pre-established exit condition (e.g., target post-grazing height or residual biomass) is reached.

Rest period is the interval, also expressed in days, between the animals’ exit from the paddock and their subsequent return, allowing for pasture recovery and regrowth.

Comments 9: Conclusions are appropriate and accurate.

Response 9: We appreciate the feedback

Reviewer 4 Report

Comments and Suggestions for Authors

Thank you for the paper you sent. Here are my comments on it:
- First of all, in my opinion, the article lacked reference to a control group, there was no “normal” pasture for these animals.
- Lines 88- 89 Do these months correspond to the months of whole grazing?
- Figure 1 Please correct the chart captions into English (legend and month names).
- lines 147- 148 I don't quite understand the part about “variable stocking rate”- nowhere later in the text is this referred to, what is this variability. The second question is, was the determinant of the end of the grazing period the achievement of a body weight of 32 kg? If so, shouldn't it also be indicated how long it took the animals on the different types of cultivars to reach these weights?
- Table 2 No units are given.
- lines 174- 178 There is a lack of information regarding at which period of this grazing the observations were made (right after the animals were released, in the middle of the experiment, or at the end?). This may affect the variation in behavior. In addition, it is not described how many people conducted observations, were they conducted at the same time for all cultivars?
- In Table 5, please change the placement of the letter “a” next to “7.34”
- In Table 6. (or in the methodology) I am missing information regarding the average weights for each group.
- Also missing are photos of the pastures themselves for comparison.

Author Response

We appreciate the reviewers' recommendations for improving the article. We have made the necessary corrections to meet the requested requirements, and we have also explained why we were unable to follow some of the suggested advice.

Comments  1: Thank you for the paper you sent. Here are my comments on it:
- First of all, in my opinion, the article lacked reference to a control group, there was no “normal” pasture for these animals.

Response 1: We appreciate the reviewer’s comment. The objective of this study was to evaluate different Urochloa cultivars with the specific aim of reducing reliance on cv. Marandu monocultures in the Brazilian Northeast. Therefore, we focused on comparing these cultivars rather than including a “normal” pasture as a control group, since Marandu monocultures are already the predominant standard in the region.

Comments 2: - Lines 88- 89 Do these months correspond to the months of whole grazing?

Response 2: There was no full grazing period throughout the experiment. More information about experimental period was added to the paper

Comments 3:- Figure 1 Please correct the chart captions into English (legend and month names).

Response 3: We apologize for the error and have now fixed it.

Comments 4:- lines 147- 148 I don't quite understand the part about “variable stocking rate”- nowhere later in the text is this referred to, what is this variability. The second question is, was the determinant of the end of the grazing period the achievement of a body weight of 32 kg? If so, shouldn't it also be indicated how long it took the animals on the different types of cultivars to reach these weights?

Response 4: We thank the reviewer for this observation. The term “variable stocking rate” refers to the adjustments made to the number of animals per area based on forage availability throughout the experimental period. As described in the Materials and Methods section, stocking rate was calculated considering the relationship between the number of animal units (AU) and the occupied area over time.

The slaughter weight of 32 kg was established as the endpoint for each individual animal, but not as the determinant for ending the grazing period of the paddock or the evaluation of the treatments. To maintain a minimum of six animals per treatment throughout the experiment, when an animal reached the slaughter weight, it was replaced by another. However, these replacement animals were not included in the data analysis related to animal performance. Therefore, while the individual animal’s participation ended upon reaching 32 kg, the evaluation of each Urochloa cultivar continued, ensuring the consistency of the experimental design.

Comments 5: - Table 2 No units are given.

Response 5: We apologize for the error and have now added the units correctly.

Comments 6:- lines 174- 178 There is a lack of information regarding at which period of this grazing the observations were made (right after the animals were released, in the middle of the experiment, or at the end?). This may affect the variation in behavior. In addition, it is not described how many people conducted observations, were they conducted at the same time for all cultivars?

Response 6: We thank the reviewer for this observation. We have improved the description of the Materials and Methods section regarding the evaluation of ingestive behavior to clarify these aspects. Specifically, we included information about the timing of the observations in relation to the grazing period, as well as the fact that all evaluations were conducted simultaneously across all treatments by a trained evaluator.

Comments 7: - In Table 5, please change the placement of the letter “a” next to “7.34”

Response 7: work for necessary adjustment.

Comments 8: - In Table 6. (or in the methodology) I am missing information regarding the average weights for each group.

Response 8: The initial average body weight of the animals is provided in the Materials and Methods section, and Table 6 presents the average daily gain, which allows for the calculation of the animals' weight progression throughout the experiment. We believe that these pieces of information sufficiently characterize the productive performance of the animals in each group.

Comments 9:- Also missing are photos of the pastures themselves for comparison.

Response 9: We apologize, but we do not have simultaneous photos of the treatments.

Round 2

Reviewer 1 Report

Comments and Suggestions for Authors

Simple Summary

The text states that Ipyporã performed better than another cultivar; however, since the work addresses different cultivars, as it is currently written, I am unable to determine which cultivar it is referring to.

Methodology

Please describe that composite forage samples were prepared for chemical analysis before describing the analyses performed (Lines 139–143).

Author Response

We appreciate the feedback and apologize for any remaining errors. All comments have been carefully reviewed, and this change was made to improve clarity and better reflect the data presented.

Comments 1:  Simple Summary

The text states that Ipyporã performed better than another cultivar; however, since the work addresses different cultivars, as it is currently written, I am unable to determine which cultivar it is referring to.

Response 1: Thank you for your observation. We have revised the sentence to clarify the comparison. The text now specifies that the Ipyporã cultivar was superior to the average of the other evaluated cultivars.

Methodology

Please describe that composite forage samples were prepared for chemical analysis before describing the analyses performed (Lines 139–143).

Response: Thank you for your suggestion. We have revised the paragraph to clarify that composite forage samples were prepared prior to the chemical analyses. The revised text now presents this information in the correct order to improve clarity and methodological accuracy.

Reviewer 3 Report

Comments and Suggestions for Authors

I carefully evaluated this revised version of the manuscript, all sections were improved by the authors according to reviewers' comments, so I recommend this manuscript for publishing in the Animals Journal in the present form!

Author Response

We appreciate the feedback and apologize for any remaining errors. All comments have been carefully reviewed, and this change was made to improve clarity and better reflect the data presented.

I carefully evaluated this revised version of the manuscript, all sections were improved by the authors according to reviewers' comments, so I recommend this manuscript for publishing in the Animals Journal in the present form!

Response: We would like to thank the reviewers for their feedback. Their comments were essential to improving this work. We are grateful and satisfied that we achieved our objectives and adequately addressed the recommendations made.

Reviewer 4 Report

Comments and Suggestions for Authors

Thank you for sending the revised version of the paper. I am sending my comments:
- line 39 “forage” in lower letter
- line 40 Why was only the name of one cultivar used in the keywords?
- line 75 The word “Urochloa” please write in italic.
- Table 2 Please remove the “1” for “dry matter.”
- lines 185- 193 So for 48 hours the animals were observed by 1 person? What about the period when they were not in the pasture, this observation was continued?
- Results in general. I don't understand why an average for all cultivars was drawn in the results, when the purpose of the study was to compare them (e.g. lines 205, 232 etc.) The description of the results needs to be corrected. Perhaps it would be better to give ranges of results.
- line 220 Move the p-value after “Xaraes”, one should be consistent in describing the results.
- line 236 "were similar" not "are similar."
- line 275 "was" not "is"
- lines 424- 425 Are you sure about this?

Author Response

We appreciate the feedback and apologize for any remaining errors. All comments have been carefully reviewed, and this change was made to improve clarity and better reflect the data presented.

Comments 1: Thank you for sending the revised version of the paper. I am sending my comments:
- line 39 “forage” in lower letter

Response:  The correction was made.

- line 40 Why was only the name of one cultivar used in the keywords?

Response: We chose to include only the name of Ipypora because of its prominence in our results and because it is a cultivar that has been receiving attention in scientific studies.

- line 75 The word “Urochloa” please write in italic.

Response: The correction was made.

- Table 2 Please remove the “1” for “dry matter.”

Response: The correction was made.

- lines 185- 193 So for 48 hours the animals were observed by 1 person? What about the period when they were not in the pasture, this observation was continued?

Response: Thank you for your comment. There was more than one observer, which is why we revised the text to clarify this point. As stated in the text, the animals were observed for 48 continuous hours.

- Results in general. I don't understand why an average for all cultivars was drawn in the results, when the purpose of the study was to compare them (e.g. lines 205, 232 etc.) The description of the results needs to be corrected. Perhaps it would be better to give ranges of results.

Response: Thank you for your observation. As no statistical differences were detected among the cultivars, we chose to report the average value in the text in order to provide a general overview for the reader. Since the individual values are already presented in the table, we felt that including the full range in the text would be redundant. However, we are open to revising this section if you believe it would improve the clarity of the results.

- line 220 Move the p-value after “Xaraes”, one should be consistent in describing the results.

Response: The change was made.

- line 236 "were similar" not "are similar."

Response: The correction was made.

- line 275 "was" not "is"

Response: The correction was made.

- lines 424- 425 Are you sure about this?

Response: Thank you for your question. This conclusion was drawn based on a holistic analysis of the animals’ behavior, taking into account variables such as leaf allowance and all grazing behavior data.